# Sequence and structure comparison of ATP synthase $F_0$ subunits 6 and 8 in notothenioid fish

**Gunjan Katyal[1], Brad Ebanks[1], Magnus Lucassen[2], Chiara Papetti[3], Lisa Chakrabarti[1,4]** *

**1** School of Veterinary Medicine and Science, University of Nottingham, Sutton Bonington, United Kingdom, **2** Alfred Wegener Institute, Bremerhaven, Germany, **3** Biology Department, University of Padova, Padova, Italy, **4** MRC-Versus Arthritis Centre for Musculoskeletal Ageing Research, Nottingham, United Kingdom

* lisa.chakrabarti@nottingham.ac.uk

**Data Availability Statement:** All relevant data are within the manuscript and its Supporting information files.

**Funding:** Gunjan Katyal was supported by Vice Chancellor's International Scholarship for Research

## Abstract

Mitochondrial changes such as tight coupling of the mitochondria have facilitated sustained oxygen and respiratory activity in haemoglobin-less icefish of the Channichthyidae family. We aimed to characterise features in the sequence and structure of the proteins directly involved in proton transport, which have potential physiological implications. ATP synthase subunit a (*ATP6*) and subunit 8 (*ATP8*) are proteins that function as part of the $F_0$ component (proton pump) of the $F_0F_1$ complex. Both proteins are encoded by the mitochondrial genome and involved in oxidative phosphorylation. To explore mitochondrial sequence variation for ATP6 and ATP8 we analysed sequences from *C. gunnari* and *C. rastrospinosus* and compared them with their closely related red-blooded species and eight other vertebrate species. Our comparison of the amino acid sequence of these proteins reveals important differences that could underlie aspects of the unique physiology of the icefish. In this study we find that changes in the sequence of subunit a of the icefish *C. gunnari* at position 35 where there is a hydrophobic alanine which is not seen in the other notothenioids we analysed. An amino acid change of this type is significant since it may have a structural impact. The biology of the haemoglobin-less icefish is necessarily unique and any insights about these animals will help to generate a better overall understanding of important physiological pathways.

## Introduction

The oceans which surround Antarctica, and their sub-zero temperatures provide a home to fish of the suborder Notothenioidei—a prime example of a marine species flock.

Notothenioids are renowned for their physiological adaptations to cold temperatures. This includes the ability to synthesise antifreeze glycoproteins (AFGP) and antifreeze-potentiating proteins (AFPP) [1]. The capacity to synthesise antifreeze glycopeptides (AFGPs) is a biochemical adaptation that enabled the Notothenioidei to colonize and thrive in the extreme polar environment [2]. These proteins are largely composed of a Thr-Ala-Ala repeat with a

Excellence, University of Nottingham (2018-2021) https://www.nottingham.ac.uk/. Brad E Banks was supported by the Biotechnology and Biological Sciences Research Council https://bbsrc.ukri.org/ [grant number BB/J014508/1. The funders had no role in study design, data collection and analysis, decision to publish, or preparation of the manuscript.

**Competing interests:** The authors have declared that no competing interests exist.

conjugated disaccharide via the hydroxyl group of the Thr residue and reduce the freezing point of the animals internal fluids [3,4].

Channichthyidae, contained within the Notothenioid suborder, are remarkable due to the absence of haemoglobin and, in some species, myoglobin too [5–7]. The sub-zero temperatures of the water they inhabit allow the highest levels of oxygen solubility, which is suggested to facilitate their survival despite the loss of globin proteins [7].

Myoglobin is absent in the oxidative skeletal muscle in all icefish, but the absence of myoglobin in cardiac muscle has been reported in only six of the species of the Channichthyinae [8,9]. While the molecular genetics of how myoglobin expression has been lost have been studied, the physiological differences between those that express and those that do not express myoglobin are not fully understood. Small intracellular diffusion distances to mitochondria and a greater percentage of cell volume occupied by mitochondria are two evolutionary adaptations that might compensate for the absence of myoglobin [10,11]. In the particular case of *Champsocephalus gunnari*, the mRNA transcript of myoglobin is present in the cardiac tissue but a 5-bp frameshift insertion hinders the synthesis of protein from the mRNA transcript [8,12].

Notothenioidei have high densities of mitochondria in muscle cells, versatility in mitochondrial biogenesis and a unique lipidomic profile [13–15]. These features have also been hypothesised to facilitate sustained oxygen consumption and respiratory activity in the absence of haemoglobin and myoglobin.

Complex V of the electron transport chain, ATP synthase, is responsible for the production of intracellular ATP from ADP and inorganic phosphate. Composed of an F$_0$ and F$_1$ component, the F$_0$ component is responsible for channelling protons from the intermembrane space across the inner mitochondrial membrane and into the mitochondrial matrix [16–18]. The rotation of the c-ring in F$_0$, and with this the γ-subunit of the central stalk, facilitates the translocation of protons across the inner mitochondrial membrane that ultimately drives the catalytic mechanism of the F$_1$ component [19,20].

The motor unit F$_0$, embedded in the inner membrane of mitochondria, is composed of subunits b, OSCP (oligomycin sensitivity conferring protein), d, e, f, g, h, i/j, k which are encoded by nuclear genes and subunits a (ATP6) and 8 (ATP8), which are encoded by mitochondrial genes [21]. Despite the structure of the complex having been first resolved decades ago, and hypotheses of the chemical mechanism were developed over half a century ago, significant breakthroughs continue to be made in our understanding of both the structure and function of the enzyme and its F$_0$ component [22–25].

Both ATP synthase subunit a (ATP6) and subunit 8 (ATP8) are proteins that function as part of the F$_0$ component of ATP synthase, encoded by genes that overlap within the mitochondrial genome [26]. This overlap is over a short, but variable between species, base pair sequence where the translation initiation site of subunit 8 is contained within the coding region of subunit 6.

The peripheral stalk is a crucial component of the F$_0$ component forming a physical connection between the membrane sector of the complex and the catalytic core. It provides flexibility, aids in the assembly and stability of the complex, and forms the dimerization interface between ATP synthase pairs [27]. ATP8 is an integral transmembrane component of the peripheral stalk, serving an important role in the assembly of the complex [28]. The C-terminus of ATP8 extends 70 Å from the surface of the makes contacts with subunits b, d and F$_6$, while the N-terminus has been reported to make connections with subunits b, f and 6 in the intermembrane space [29,30]. Subunit 8 is also known to play a role in the activity of the enzyme complex [31].

ATP6 is an α-helical protein embedded within the inner mitochondrial membrane and it interacts closely with the c-ring of $F_0$, providing aqueous half-channels that shuttle protons to and from the rotating c-ring [17,32]. It has previously been reported that ATP6 has at least five hydrophobic transmembrane spanning α helices domain, where two of the helices h4 and h5 are well conserved across many species [33].

Proteins coded by mitochondrial DNA (mtDNA) are involved in oxidative phosphorylation and can directly influence the metabolic performance of this pathway. Evaluating the selective pressures acting on these proteins can provide insights in their evolution, where mutations in the mtDNA can be favourable, neutral, or harmful. The amino acid changes can cause inefficiencies in the electron transfer chain, causing oxidative damage by excess production of reactive oxygen species and eventually interrupting the production of mitochondrial energy. Due to the tight coupling of icefish mitochondria relative to their red-blooded relatives, any changes in the structure of ATP Synthase subunits, particularly those directly involved in the transport of protons across the membrane, could result in significant physiological outcomes [34].

In this work, we combine sequence analyses and secondary structure prediction analyses to explore mitochondrial genetic variation for ATP6 and ATP8 in the Notothenioidei suborder species as well as other vertebrate species. The species considered include *Champsocephalus gunnari*, *Chionodraco rastrospinosus* and *Chaenocephalus aceratus* from the Channichthyidae family, *Notothenia coriiceps* and *Trematomus bernacchii* from the Nototheniidae family and the sub-Antarctic *Eleginops maclovinus* from family Eleginopsidae, all the broader Notothenioidei suborder. The species of suborder Notothenioidei are further compared with the following eight vertebrates: *Homo sapiens* (family: Hominidae), *Nothobranchius furzeri* (family: Nothobranchiidae), *Danio rerio* (family: Cyprinidae), *Anolis carolinensis* (family: Dactyloidae), *Cavia porcellus* (family: Caviidae), *Balaena mysticetus* (family: Balaenidae), *Heterocephalus glaber* (family: Heterocephalidae), and Lasiurus *borealis* (family: Vespertilionidae) to shed light on the changes of these proteins in the notothenioid species by comparing them to better characterised diverse vertebrate species. These species choices help us decipher amino acid changes specific to notothenioids and those that are potentially species specific (S1 Fig).

## Methodology

### Extraction of gene and protein sequences of ATP8 and ATP6 suborder Notothenioidei and other vertebrates

The list of complete coding sequences (CDS) and protein sequences of the proteins were obtained from the National Centre for Biotechnology Information (NCBI) protein database search, we chose only the Refseq (provides a comprehensive, integrated, non-redundant, well-annotated set of sequences, including genomic DNA, transcripts, and proteins) sequence queries (https://www.ncbi.nlm.nih.gov/ lMSast searched:17th August 2020). Though these sequences have been taken from highly reliable Refseq database [35] validated by different sources it is important to recognise they could still be prone to error.

### Multiple protein sequence alignment (MSA)

(-/-) indicates absence of both haemoglobin and myoglobin genes, whereas (-/+) indicate absence of haemoglobin but presence of myoglobin. The sequences for the Notothenioidei suborder species *C. gunnari* (-/-), *C. rastrospinosus* (-/+), *C. aceratus* (-/-), *N. coriiceps* (+/+), *T. bernacchii*, *E. maclovinus* (+/+), and eight other vertebrate species, *N. furzeri*, *D. rerio*, *A. carolinensis*, *C. porcellus*, *B. mysticetus*, *H. glaber*, *L. borealis*, *H. sapiens* were aligned using Clustal

omega [36] to prepare the initial alignment of ATP6 protein under the criteria of the presence and the absence of haemoglobin and myoglobin proteins in the species, the alignments were also verified using the other two progressive methods, MAFFT [37] and MUSCLE [36]. The same method was applied for protein ATP8. The MSA was visualised and edited using JAL-VIEW [38]. The eight vertebrate species were selected as well known and sequenced representative of different groups under vertebrate: fish (*N. furzeri* and *D. rerio*), reptiles (*A. carolinesis*), mammals (*C. porcellus*, *H. glaber*, *L. borealis*, *H. sapiens*, *B. mysticetus*). *H.sapiens* sequences have been included in our analyses since much of what is known about these proteins has previously been characterised in humans. The selection of these different species shows the conservation of these mitochondrial proteins across vertebrate species, including *H. sapiens*.

## Codon alignment

Complete nucleotide coding sequences for genes *ATP6* and *ATP8* from the fourteen vertebrate species were retrieved from NCBI GenBank database (see Table 1). The sequences were aligned using Clustal omega [36] and were manually edited and visualised as codons using MATLAB version R2018b (9.5.0).

## Comparison of properties of amino acids among the sequence from the above-mentioned species

Using the ExPASy [39] tool ProtScale [40], different amino acid properties such as the molecular weight of amino acids across the sequence, hydrophobicity trend of amino acids, α—helix forming amino acids, average flexibility trend and mutability for the protein ATP6 were compared graphically among the seven fish species (5 Antarctic, 1 sub-Antarctic, *D. rerio* and *N. furzeri*) (https://web.expasy.org/protscale/).

## Structure prediction for protein sequences

The MSA was structurally validated using the structure prediction tool I-TASSER [41] (Iterative Threading ASSEmbly Refinement) a hierarchical approach to protein structure and function prediction, to generate the protein structure for AT6 from different species (https://zhanglab.ccmb.med.umich.edu/I-TASSER/). The structures were validated using SAVES v6.0 (https://saves.mbi.ucla.edu/), using ERRAT [42], PROCHECK [43,44] and ProSA-web [45]. (Figures in supplementary files).

## Figures

Protein structure images were produced with PyMOL v. 2.3.2. (The PyMOL Molecular Graphics System, Version 2.0 Schrödinger, LLC.) Graphs were produced with MATLAB version R2018b (9.5.0). Sequence logos were created using the webserver WebLogo using alignment of 5947 vertebrate (NCBI:txid7742) protein sequences for the protein ATP6 (http://weblogo.threeplusone.com/). Using RefSeq sequences with custom range of sequence length of 224–231 to obtain full sequences only (searched: 3$^{rd}$ May 2021).

## Results

### Codon alignment

MSA of all the sequences of ATP8 (see Fig 1) and ATP6 (see Fig 2) from the different vertebrate species (see Table 1) for both nucleotide (codon) and proteins identified several

**Table 1. Features of nucleotide and protein sequences for ATP synthase F$_0$ subunit 6 and 8.**

| Species/Features | C. gunnari | C. rastrospinosus | C. aceratus | N. coriiceps | T. bernacchii | E. maclovinus | N. furzeri | D. rerio | A. carolinensis | L. borealis | H. glaber | C. porcellus | B. mysticetus | H. sapiens |
|---|---|---|---|---|---|---|---|---|---|---|---|---|---|---|
| Common Name | Mackerel Icefish | Ocellated icefish | Blackfin Icefish | Marbled rockcod | Emerald rockcod | Rockcod | Killifish | Zebra fish | Lizard | Eastern red bat | Naked mole rat | Guinea Pig | Bowhead Whale | Humans |
| Accession No. ATP6 (protein) | YP_006575887.1 | YP_009519992.1 | AEH05456.1 | BBC27483.1 | ANN44664.1 | YP_009340798.1 | YP_002456261.1 | NP_059336.1 | ACD81888.2 | YP_005255233.1 | YP_004222617.1 | QIQ22938.1 | AWM99473.1 | YP_003024031.1 |
| Accession No. ATP6 (nucleotide) | NC_018340.1 | NC_039543.1 | NC_015654.1 | NC_015653.1 | KU166863 | NC_033386.1 | NC_011814.1 | NC_002333.2 | NC_016873.1 | NC_001573.1 | NC_015112.1 | NC_000884.1 | NC_005268.1 | NC_012920.1 |
| Accession No. ATP8 (protein) | YP_006575886.1 | YP_009519991.1 | YP_004581501.1 | YP_004581488.1 | ANN44663.1 | YP_009340797.1 | YP_002456260.1 | NP_059335.1 | ACD81887.2 | YP_005255232.1 | YP_004222616.1 | NP_008755.1 | NP_944611.1 | NC_012920.1 |
| Haemoglobin | - | - | - | + | + | + | + | + | + | + | + | + | + | + |
| Myoglobin | - | + | - | + | + | + | + | + | + | + | + | + | + | + |
| Length of nucleotide ATP6 | 683 | 695 | 695 | 695 | 695 | 695 | 682 | 683 | 680 | 683 | 680 | 680 | 680 | 680 |
| 5' flanking region ATP8 | 74nt | 75nt | 75nt | 75nt | 75nt | 75nt | 74nt | 73nt | 67nt-trn-Lysine | 71nt | 73nt | 68nt | 71nt | 71nt |
| ATP6-Start codon | atg | gtg | gtg | gtg | atg | gtg | atg | atg | atg | atg | atg | atg | atg | atg |
| 4_nucleotides at 5'end | - | + GTG-AAC-CTG-ACC | + GTG-AAC-CTG-ACC | + GTG-GTC-CTG-ACC | + ATG-AAC-TTG-GCC | + GTG-AAC-CTG-ACC | - | - | - | - | - | - | - | - |
| 4_amino acid at 5'end | - | +MNLT | +MNLT | +MVLT | +MNLA | +MNLT | - | - | - | - | - | - | - | - |
| Codon aligning at position 35/39 (nucleotide) | GCT | TCT | TCT | TCT | TCC | TCT | CTT | ACA | AAT | ACC | CCC | CCC | CCA | CCA |
| Residue Aligning at position 35/39 (protein) | Alanine | Serine | Serine | Serine | Serine | Serine | Leucine | Threonine | Asparagine | | | | | |
| Residues at positions 38-39 aligned to 42-43 residues | Valine-Isoleucine | Valine-Isoleucine | Valine-Isoleucine | Valine-Isoleucine | Valine-Valine | Valine-Valine | Tryptophan-Leucine | Tryptophan-Isoleucine | Leucine-Valine | Isoleucine-Asparagine | Isoleucine-Asparagine | Isoleucine-Asparagine | Isoleucine-Asparagine | Isoleucine-Asparagine |
| Properties of substitution | Non-Polar | Non-Polar | Non-Polar | Non-Polar | Non-Polar | Non-Polar | Non-polar aromatic AA -Hydrophobic branched AA | Non-polar aromatic AA -Hydrophobic branched AA | Hydrophobic AA | Hydrophobic AA —Polar, non-charged AA | Hydrophobic AA —Polar, non-charged AA | Hydrophobic AA—Polar, non-charged AA | Hydrophobic AA—Polar, non-charged AA | Hydrophobic AA —Polar, non-charged AA |
| Structural change at position 38-39 aligned to 42-43 residues | strand-strand | coil-coil | coil-coil | coil-coil | strand-strand | strand-strand | strand-strand | strand-strand | strand-strand | coil-coil | coil-coil | coil-coil | coil-coil | coil-coil |

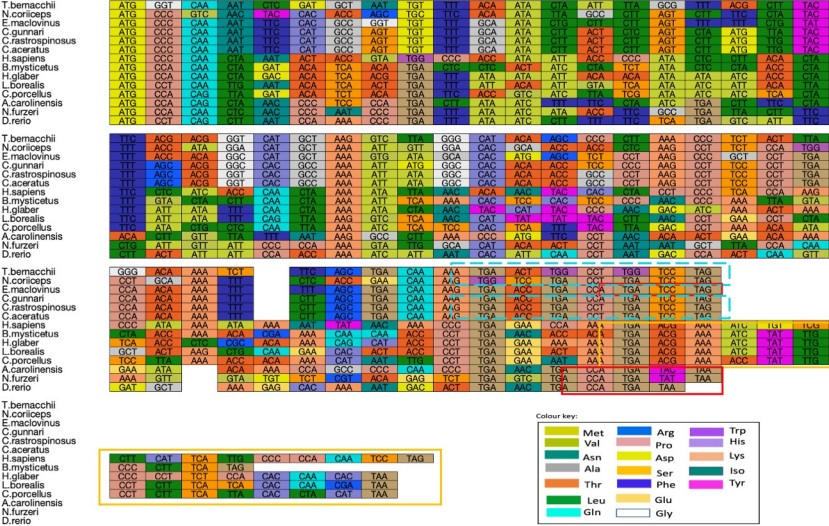

**Fig 1. Multiple sequence alignment for nucleotide sequences of ATP synthase subunit 8.** Multiple codon alignment of nucleotide sequences of ATP synthase subunit 8 was created using the Clustal omega alignment of nucleotides we screened five Antarctic and one sub-Antarctic fish species and eight vertebrate outgroups same as ATP6 MSA (See Fig 1 for colour key). The highlighted boxes show the overlap of the ATP8 and ATP6 sequences for different species where different colour of the boxes correspond to the different lengths of overlap.

conserved codons and amino acid residues. The sequence knowledge was gathered from curated entries in RefSeq which nevertheless could be subject to error.

Five of the six Antarctic fish species have twelve nucleotides (four codons) at the 5' end of the gene sequence which are not found in the other eight vertebrate species. The codon alignment ATP6 for species *E. maclovinus*, *N. coriiceps*, *C. rastrospinosus* and *C. aceratus* show that GTG codes for methionine, as the start codon for the protein. GTG which is originally known for coding the amino acid valine has been accepted as a mitochondrion start codon for invertebrate mitogenomes [46–48]. A common feature with the species that have GTG as a start codon is that *N. coriiceps*, *E. maclovinus*, *C. rastrospinosus* have genes coding for myoglobin, where the latter is devoid of haemoglobin. *C. aceratus* do not express myoglobin due to a 15 bp sequence insertion, other than that difference, their myoglobin gene sequence is identical to that of *C. rastrospinosus* [9]. The only exception to this is the red-blooded species *T. bernacchii*, but this may be attributed to the unverified source of its sequence submission.

Another trend that has been observed through sequence alignment is that the species that are more similar and have the same amino acid for a particular position also have codons with the same nucleotide (nt) at the third position. 'TGA' codons or 'stop codons' are found within the translated sequence, here these code for tryptophan, as seen in human and yeast mitochondria [49]. A variation in the length of the sequences was observed, with an average length for *ATP6* nt sequence of 683 and 74 nt for *ATP8* gene sequences. The *ATP6* sequence ends with a TAA stop codon in all species except the two red blooded Antarctic fish species, *N. coriiceps* and *E. maclovinus*.

## Overlapping genes

The overlap between genes is encoded on the same strand (Table 1). The length of overlap was 22 nt in ATP8-ATP6 for the five of the six species of Notothenioidei suborder, that is excluding

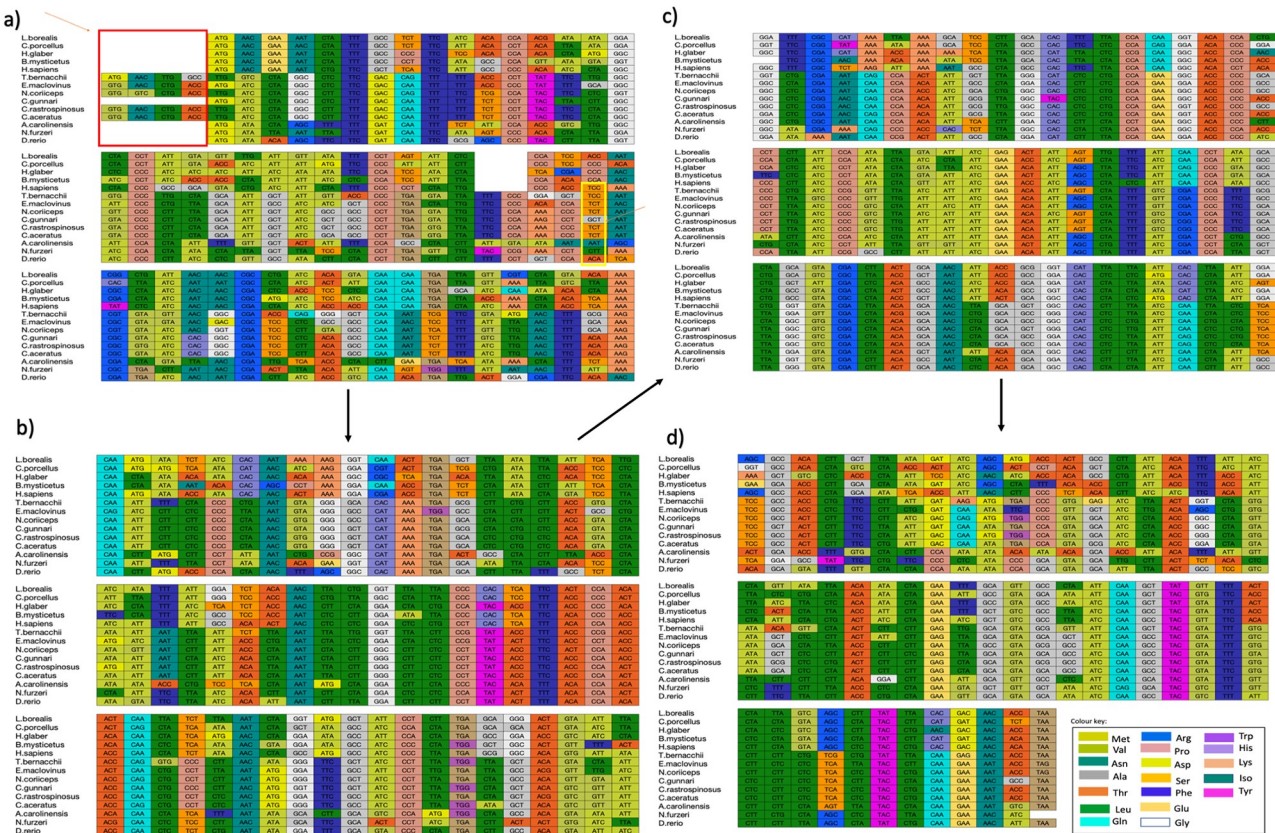

**Fig 2. (a-d) Multiple sequence alignment for nucleotide sequences of ATP synthase subunit 6.** Multiple codon alignment of nucleotide sequences of ATP synthase subunit 6 was created using the Clustal omega alignment of nucleotides for five Antarctic and one sub-Antarctic fish species and eight vertebrate outgroups and visualised) using MATLAB. The colour of the codon boxes corresponds to the respective amino acid (See colour key).

icefish *C. gunnari* where the overlap was of 10nt. Species *H. sapiens*, *H. glaber*, *L. borealis* and *C. porcellus* had an overlap of 43nt between ATP6 and ATP8. The shortest overlap between the two genes were observed in the species *A. carolinesis* has an overlap of 10nt and *N. furzeri* and *D. rerio*, have an overlap of 7nts.

## Protein alignment and structural changes in ATP6

The complete amino acid sequences for ATP8 and ATP6 were aligned separately for the fourteen vertebrate species (see Figs 3 & 4). Protein sequence alignment showed conserved residues across the species based on identity and similarity. Four Antarctic fish species, *N. coriiceps*, *T. bernacchii*, *C. rastrospinosus*, *C. aceratus* and the sub-Antarctic *E. maclovinus* have four amino acids at the N-terminal with a total of 231 residues. As previously mentioned, the only exception to this, is the species *C. gunnari* with 227 residues similar to that of other fish species, *N. furzeri* and *D. rerio*. Species *H. sapiens*, *A. carolinesis*, *L. borealis*, *H. glaber* and *C. porcellus* have 226 residues and *B. mysticetus* has 225 residues. The protein ATP6 in vertebrates is known to have 226–228 residues. In humans, four point mutations in the ATP6 gene account for 82% of disease associated with this gene, suggesting point mutations could have physiological relevance [50,51]. Common features in all fourteen species were as follows: (1) several hydrophobic amino acids (light pink) were observed to be conserved across the sequences in

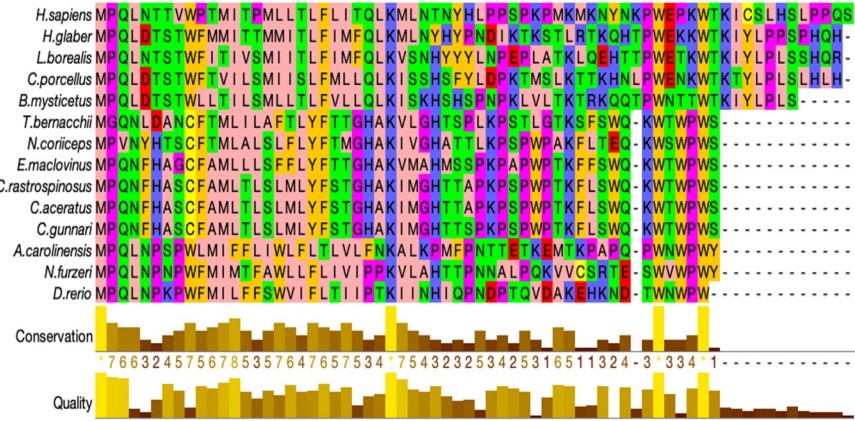

**Fig 3. Multiple sequence alignment of ATP8 protein sequences.** The ATP8 protein sequences were aligned using Clustal omega and edited using zappo colour scheme in JalView. Notothenioidei are grouped together in blue; all species are displayed to the colour corresponding to their phylogenetic closeness. (Colours according to physio-chemical properties of amino acids; Aliphatic/hydrophobic-A, I, L, M, V- light pink; Aromatic-F, W, Y- mustard; Conformationally special- Glycine, P- magenta; C-yellow; Hydrophilic- N, Q, S, Q, T- light green; Negatively charged/ D,E-Red; Positively charged/R,H,K-Blue) in jalview. The bar-graphs below represent a quantitative measure of conservation at each position. The figure was created using JalView.

the species, (2) insertions and deletions of amino acids occurred more frequently near N-termini, and (3) the C-terminal of the protein sequence is hydrophilic. Dashes in the amino acid sequence represent gaps which may be an insertion or deletion of a residue. The gap in the alignment is observed for the species *H. sapiens*, *L. borealis*, *C. porcellus*, *B. mysticetus* and *H. glaber* at position 35, and at the C-terminal end for *A. carolinesis* and *B. mysticetus*, at position 226 and 225 respectively.

The amino acid at position 35 has predominantly hydrophilic residues except in the two species *C. gunnari* and *N. furzeri*, where it is substituted with alanine or leucine respectively.

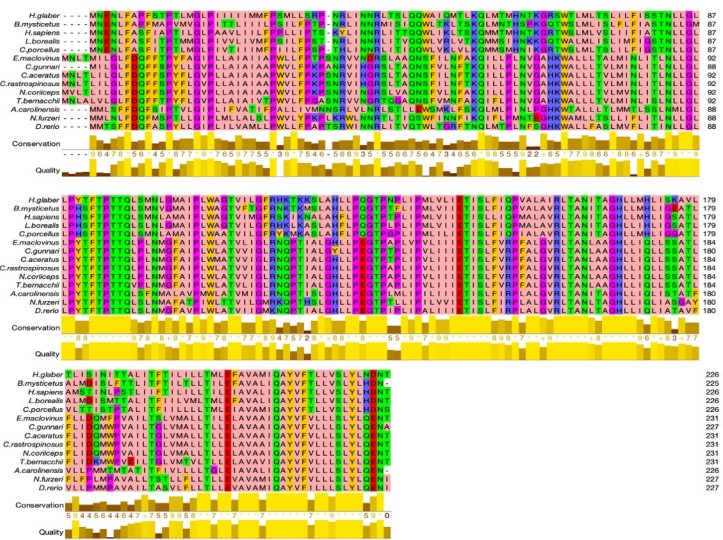

**Fig 4. Multiple sequence alignment for protein sequences of ATP synthase F$_0$ subunit 6.** The ATP6 protein sequences were aligned using Clustal omega and edited using zappo colour scheme.

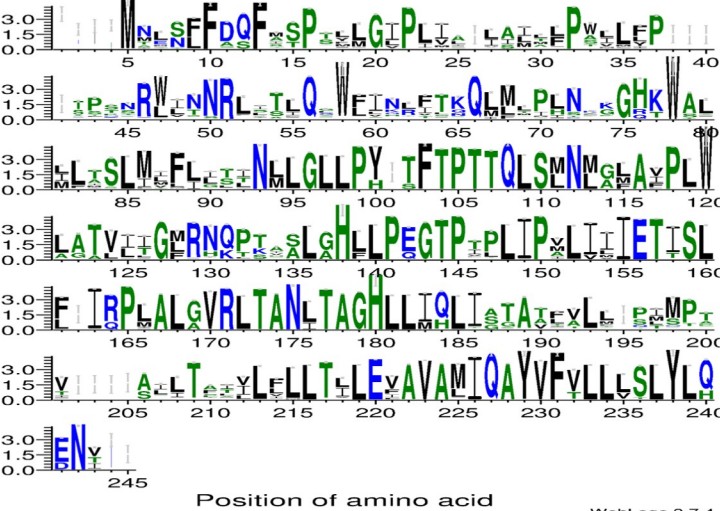

**Fig 5. Sequence logos displaying conservation of residues created for all aligned blocks of the MSA for protein ATP synthase F$_0$ subunit 6 for 5947 vertebrate species from NCBI using webserver WebLogo (http://weblogo. threeplusone.com/) the y axis represents probability of the residue occurring at that position from the MSA.**

All the Antarctic species except *C. gunnari*, the sub-Antarctic species, *E. maclovinus* and surprisingly *H. sapiens* from the mammalian species have a serine at this position. When we look at the codon alignment of the ATP6 gene, serine is encoded by codon TCT predominantly at position 39 for all the species except *T. bernacchii* and *H. sapiens* and the alanine for the species *C. gunnari* is encoded by GCT (see Fig 2).

The logo (see Fig 5) displays the conserved amino acids in the protein ATP6 for a particular position for 5947 vertebrate species. The protein is overall very conserved in the vertebrates, and position 38–39 show conservation for amino acids serine and threonine as also seen in the Antarctic species (except *C. gunnari*) and *E. maclovinus*.

A similar pattern was found in the amino acid alignment of ATP8, where the species, *H. sapiens*, *B. mysticetus*, *H. glaber*, *C. porcellus* and *L. borealis*, that showed a gap in the previous alignment have hydrophilic residues whereas the other species have a gap at the position 47. This observation could be attributed to the overlapping nature of the nucleotide sequences coding for the two proteins. The protein sequence of ATP6 was observed to be more conserved than ATP8. The amino acid sequences at the N- terminal are more diverse, and the methionine residues are usually followed by amino acids with short polar side chains [52]. Alanine is a non-polar amino acid whereas serine is a polar amino acid. The hydrophobicity plot, average flexibility, mutability, and coil prediction across the sequences has shown that *T. bernacchii* and *E. maclovinus* show similar trends in their physico-chemical properties across the sequence. *Nototentia coriiceps*, *C. aceratus* and *C. rastrospinosus* follow this trend. *Champsocephalus gunnari* is the only species out of the seven fish species compared, that is different from the others (see Fig 6).

Protein structure differences were predicted at position 38–39 for species *C. gunnari* (icefish), *N. furzeri*, *D. rerio* and *A. carolinesis*, where a strand-strand structure is found at that position. All other species have coil structures at those positions (see Fig 7). For species *T. bernacchii* and *E. maclovinus* there is also a prediction for a strand structure at positions 42–43.

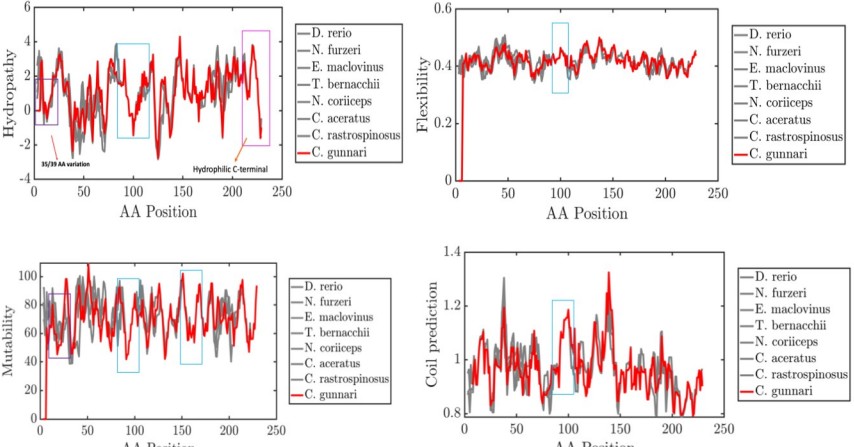

**Fig 6. Primary sequence features of ATP Synthase F$_0$ subunit 6 in species *C. gunnari* (red), *C. rastrospinosus*, *C. aceratus*, *N. coriiceps*, *T. bernacchii*, *E. maclovinus*, *N. furzeri* and *D. rerio*.** Red Box: N-terminal property changes, Purple Box: Changes in properties observed at 35/39 variation, blue box: Conserved regions 90–170 (Active site 160–169), Pink Box: C-terminal low hydrophobicity. A difference in the peaks have been observed for different properties (highlighted) such as molecular weight and hydrophobicity of amino acid residues across the sequence and other properties such as tendency of amino acid residues towards beta-sheet, bulkiness and flexibility.

## Discussion

We present our analyses highlighting differences in sequence and structure observed in the two proteins of complex V, ATP8 and ATP6, encoded by mtDNA between the red- and white blooded species of suborder Notothenioidei. Our analyses are based on the current genome annotation available which is subject to change as more information becomes available. We have only selected *RefSeq* sequences as these are reviewed by NCBI and represent a compilation of the current knowledge of a gene and protein products and is synthesised using information integrated from multiple sources. *RefSeq* is used as a reference standard for a variety of purposes such as genome annotation and reporting locations of sequence variation. It is important to acknowledge however that database information is regularly updated and may change. Currently, the RefSeq and GenBank entries available for a ATP6 sequences for the Antarctic/sub-Antarctic fish, NC_015653.1, AP006021.1 (*N. coriiceps*), NC_039543.1, MF622064.1 (*C. rastrospinosus*), NC_033386.1, KY038381.1 (*E. maclovinus*), NC_015654.1, YP_004581502.1 (*C. aceratus*), which are submitted by different authors, have the start codon as GTG for the five species of Notothenioidei suborder. The protein length of ATP6 has been consistent in all the entries, 231 amino acids.

It has previously been shown that mitochondria from icefish are more tightly coupled than those of their red-blooded counterparts [34]. Mitochondria that are tightly coupled usually have competent membranes and protons can only get into the matrix by passing through complex V. The red-blooded species *N. coriiceps*, *E. maclovinus*, *T. bernacchii*, the two icefish *C. rastrospinosus* (devoid of hb, have mb), *C. aceratus* (devoid of hb, do not express mb but have a nearly identical gene to that of C. rastrospinosus for mb), have an additional 12 nucleotides at the N-terminal. The only exception to this is the icefish *C. gunnari* which is completely devoid of both hb and mb. Since *C. gunnari* is the extreme of all the species of Notothenioidei suborder in question in terms of loss of globins, the change observed could be an altered variation for the gene.

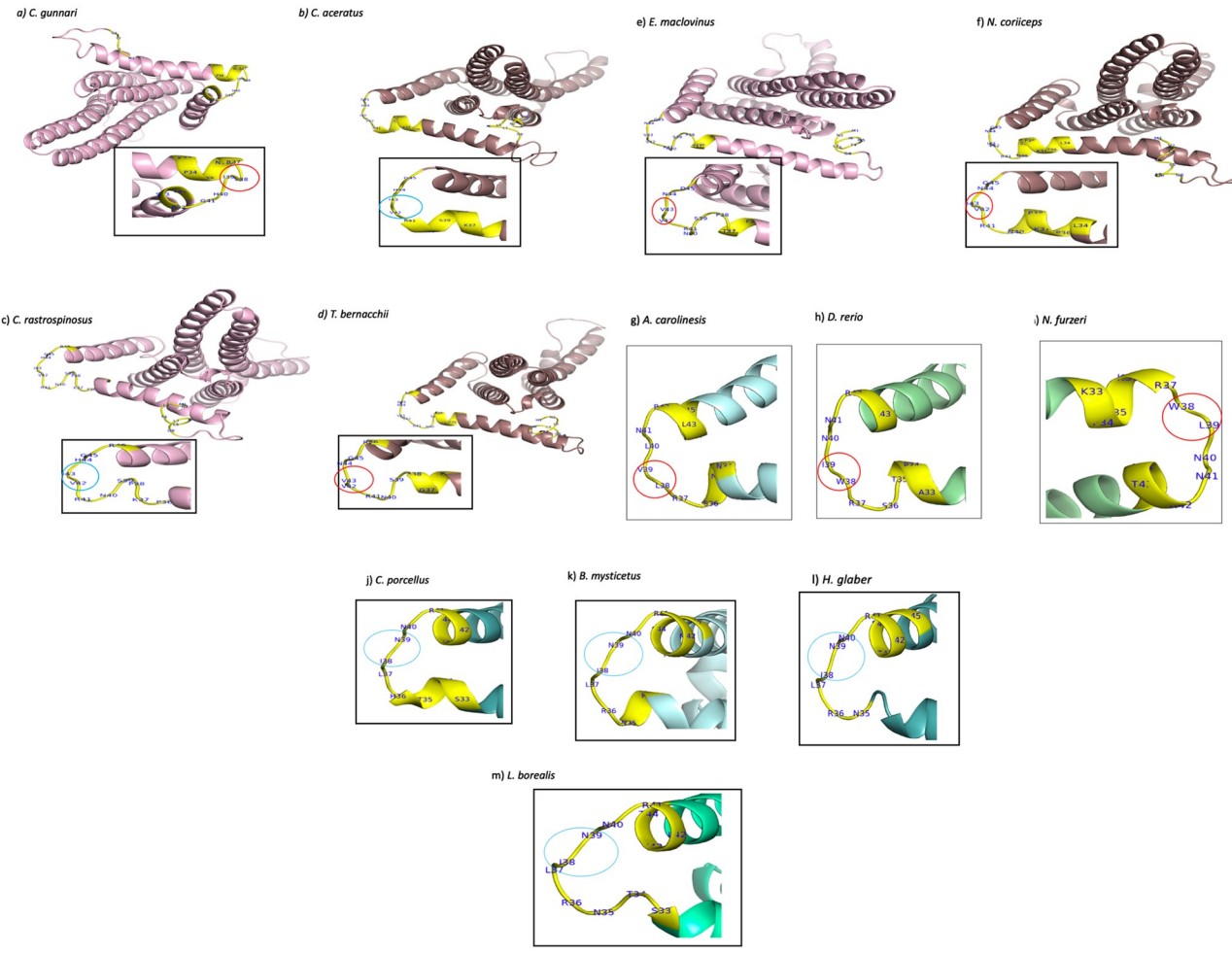

**Fig 7. Representative structures of ATP synthase F$_0$ subunit 6 for the fourteen vertebrate species created using I-TASSER [41] suite and visualised and edited using PyMOL v. 2.3.2.** a) *C. gunnari*(-/-) residues 38 (valine) and 39 (isoleucine) shows strand structure b) *C. aceratus*(-/-) residues 42(valine) and 43 coil (isoleucine), aligning with 38/39 in MSA, show a coil structure c) *C. rastrospinosus*(-/+) residues 42-Valine and 43-Isoleucine has a coil structure d) *T. bernacchii*(+/+) residues 42-Valine and 43-Valine show a strand structure e) *E. maclovinus* (+/+) residues 42 -Valine and 43-Valine show a strand structure f) *N. coriiceps*(+/+) residues 42 (Valine) 43 (isoleucine) has a coil structure. g) *A. carolinesis* residues 38 (Leucine) and 39(Valine) show a strand structure h) *D. rerio* residues 38 (tryptophan) and 39(Isoleucine) show a strand structure i) *N. furzeri* residues 38 (Tryptophan) and 39 (Leucine) show a strand structure. j) *C. porcellus* residues 38 (Isoleucine) and 39 (Asparagine) show a coil structure k) *B. mysticetus* residues 38 (Isoleucine) and 39 (Asparagine) show a coil structure. l) *H. glaber* residues 38 (Isoleucine) and 39 (Asparagine) show a coil structure—m) *L. borealis* residues 38 (Isoleucine) and 39 (Asparagine) show a coil structure.

## GTG as an alternative start codon

The biosynthesis of proteins encoded by their respective mRNA requires an initiation codon for their translation. ATG is the usual initiation codon but GTG has been reported as initiation codon in some lower organisms, the frequency of annotated alternate codon in higher organisms is found to be less than 1% [53]. An *in-vitro* study of GTG-mediated translation of enhanced green fluorescent protein suggested that initiation with GTG codon regulates expression of lower levels of the protein and a similar observation was made for the protein endopin 2B-2 [54]. It has also been observed in a few human diseases that a mutation of the ATG initiation codon to a GTG are associated with diseases such as beta-thalassemia and Norrie disease, where GTG mutation leads to inactivation of the gene [55,56]. Another example is

a disruption caused by GTG as the initiation codon in the gene CYP2C19, which resulted in poor metabolism of a drug, mephenytoin, when compared to the gene with an ATG initiation codon [57]. Numerous studies on bacteria and lower organisms show GTG as a start codon, where the non-methionine codon is initially coded for, however, when they act as a start codon the initial amino acid is substituted with a methionine [54,58]. There is only a single report of a vertebrate species, rat, where GTG is the start codon in mtDNA [59]. An ATG to GTG exchange in human gene *FRMD7* (FERM Domain Containing 7) has been found as a first base transversion of the start codon that accounts for a mutation, causing morphological changes in the optic nerve head [60]. The level of corresponding protein expression has been shown to be lower when initiated using an alternative codon such as GTG rather than ATG [54,61]. GTG was observed as a start codon for ATP8 in fish *Philomycus bilineatus*, which adds onto the show GTG as an acceptable start codon [62].

A few but increasing number of mammalian genes have been found to give rise to an alternative initiation codon in regulatory proteins such as transcription factors, growth factors and a few kinases in humans and rats. The finding in all these studies have shown a similar trend of a lower level of protein production when compared to an ATG start codon [63–65]. It has been shown that the fish inhabiting colder climates had undergone stronger selective constraints in order to avoid deleterious mutations [66,67]. MtDNA coding genes such as ATP6, could be placed under selective pressures by low environmental temperatures. A larger ratio of substitution for different sites could indicate proteins undergoing adaptations [68]. A decrease in ATP6 activity previously reported, shows incomplete ATPase complexes that are capable of ATP hydrolysis but not ATP synthesis. ATPase complexes completely lacking subunit a, were capable of maintaining structural interactions between F1 and F0 parts of the enzyme but the interactions were found to be weaker [69].

The GTG initiation for protein ATP6 in these fish species could suggest a common parallel evolution of the translation machinery. The favouring of GTG as a start codon could also mean a higher stability of the protein as GC base pair has higher thermal stability when compared to the AT base pair which is attributed from stronger stacking interaction between GC bases and a presence of triple bond compared to that of AT double bond [70].

## Overlap of ATP8 and ATP6 genes

Protein coding genes ATP8 and ATP6 are located adjacent to each other and are overlapping on the same strand in humans and other vertebrates, with an overlap of 44 nt (NCBI: NC_012920.1) observed in the humans for the gene. It has been previously reported that ATP8-ATP6 overlap is generally of 10 nt in the fish genome [71]. Species *T. bernacchii*, *E. maclovinus*, *N. coriiceps*, *C. rastrospinosus* and *C. aceratus* show an overlap of 22 nts and *C. gunnari* has a 10 nt overlap, as reported previously in other fish genomes mentioned above. The overlap for the four out of six species of suborder Notothenioidei start from the third nucleotide for codon AAG coding for amino acid lysine whereas for the other two species, *T. bernacchii* and *C. gunnari*, it is encoded by AAA. It is hypothesised that overlaps are a mechanism for reduction of genome size and regulation of gene expression [72,73], which is seen in the species *C. gunnari* and the eight vertebrate outgroups.

The gene coding ATP8 ends with the stop codon TAG for all species of suborder Notothenioidei and TAA for the other vertebrate species, a single exception to this was *H. glaber* that ends with a TAG stop codon. It has been previously hypothesised that TAG is a sub-optimal stop codon which is less likely to be selected. A study showed that the protein encoding genes that end with TAA stop codons are, on average more abundant than those with genes ending with TGA or TAG and further shows that a switch of stop codon TAG from TGA might pass

through the mutational path of TAA stop codon which could be subject to positive selection in several groups [74].

## Protein alignment and structural changes in ATP6

The four Antarctic fish species, *N. coriiceps*, *T. bernacchii*, *C. rastrospinosus*, *C. aceratus* and the sub-Antarctic *E. maclovinus* have four amino acids at the N-terminal of ATP6 and a total of 231 residues. As previously mentioned, the only exception to this is the species *C. gunnari* with 227 residues similar to *N. furzeri* and *D. rerio*. N-terminal addition of amino acids can influence the properties of the protein, as it can change the molecular weight of the protein, the charge, hydrophobicity, and this has been seen in the yeast meta-caspase prion protein Mca1 [75].

Amino acid position 35 is populated with predominantly hydrophilic residues, apart for the two species *C. gunnari* and *N. furzeri*, where respectively, alanine and leucine are found. All the other Antarctic fish species and *E. maclovinus* have a serine at this position. When we look at the codon alignment of the ATP6 gene, serine is encoded by codon TCT at position 39 for all the species except *T. bernacchii* (encoded by TCC) and the alanine for the species *C. gunnari* is encoded by GCT. Serine is the only amino acid that is encoded by two codon sets. A common example of a missense mutation is where the single base pair can alter the corresponding codon to a different amino acid. This base substitution even though affecting a single codon can still have a significant effect on the protein production. It has been recently discovered that serine at a highly conserved position is more often encoded in TCN fashion and will tend to substitute non-synonymously to proline and alanine, which shows that codon for which serine is coded indicate different types of selection for amino acid and its acceptable substitutions [76]. This may be suggested as a reason for the presence of hydrophobic alanine observed in *C. gunnari* at position 35.

The weblogo for protein ATP6 shows overall conservation across the sequence for the vertebrates where the C-terminal of the protein is more conserved than the N-terminal. High conservation is observed from residues 85–112 and 165–185, as also seen in our MSA for the fourteen species. The position 35 is seen to be conserved preferably for threonine or serine as in the weblogo (Fig 5).

The hydrophobicity plot, average flexibility, mutability, and coil prediction across the sequences highlights differences in the physiochemical properties across the sequence of protein ATP6 in the species *C. gunnari*.

The secondary structure of a protein is the way in which protein molecules are coiled and folded in a certain way according to the primary sequence. Beta-strands give stability to the structure of a protein, its intrinsic flexibility can sometimes return it to coil configuration in order for the protein to perform other functions. Structural changes were observed at position 38–39 for species *C. gunnari*, *N. furzeri*, *D. rerio* and *A. carolinesis*, where strand-strand structure was predicted at that position. All other species are predicted to have coil structures at those positions (Figs 6 & 7). Species *T. bernacchii* and *E. maclovinus* are predicted to have strand structures at positions 42–43.

Protein structure, dynamics and function are all interlinked and it is vital to understand the structure of a protein in relation to function to comprehend molecular processes [77]. We have used the unique biology of the icefish to gain a better understanding of the variability of ATP6 and ATP8 sequence and structure which has importance for mitochondrial function.

## Conclusions

In this study we suggest that mitochondrial encoded protein ATP6 has an alternative start codon GTG in the species of suborder Notothenioidei except for the hb-less *C. gunnari*. This

could be related to a higher thermal stability with altered expression of this protein. Another striking difference observed only in *C. gunnari* for the protein, was a substitution of hydrophilic amino acid serine (TCT) to hydrophobic amino acid alanine (GCT). This could be a base substitution for thymine to guanine at N1 position of the codon that might have a structural impact on the protein. Our predictions based on the available curated sequence data now point to the need for targeted experimentation to understand the full physiological impact of our findings.

## Supporting information

**S1 Fig. A pictographic representation of the relatedness of 'ATP6 protein' sequence for notothenioids to other species using NJ-phylogenetic tree (Clustal omega[35]) analysed by taking alignment data that shows similarity in the amino acid composition of the protein for different vertebrate species (pictures source: Wikipedia.com, human skull: Bonesclones.com, naked mole rat: Wikiwand.com, *E. maclovinus*: Scanndposters.com).**
(DOCX)

**S2 Fig. Protein structure evaluations of ATP6 for fish species C. aceratus, C. gunnari, C. rastrospinosus, E. maclovinus, N. corriceps, T. bernacchii, D. rerio and N. furzeri (A-I) using SAVES v6.0 (https://saves.mbi.ucla.edu/), using ERRAT[41], PROCHECK[42,43] and ProSA-web[44].**
(DOCX)

## Author Contributions

**Conceptualization:** Gunjan Katyal, Magnus Lucassen, Chiara Papetti, Lisa Chakrabarti.

**Data curation:** Gunjan Katyal, Brad Ebanks.

**Formal analysis:** Gunjan Katyal.

**Funding acquisition:** Lisa Chakrabarti.

**Investigation:** Gunjan Katyal.

**Methodology:** Gunjan Katyal, Lisa Chakrabarti.

**Resources:** Magnus Lucassen, Chiara Papetti.

**Supervision:** Lisa Chakrabarti.

**Writing – original draft:** Gunjan Katyal, Brad Ebanks, Lisa Chakrabarti.

**Writing – review & editing:** Gunjan Katyal, Brad Ebanks, Magnus Lucassen, Chiara Papetti, Lisa Chakrabarti.

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
