## [Decision Letter · Decision Letter 0]

8 Apr 2021

PONE-D-21-00605

Sequence and structure comparison of ATP synthase F0 subunits 6 and 8 in notothenioid fish.

PLOS ONE

Dear Dr. Chakrabarti,

Thank you for submitting your manuscript to PLOS ONE. After careful consideration, we feel that it has merit but does not fully meet PLOS ONE’s publication criteria as it currently stands. Therefore, we invite you to submit a revised version of the manuscript that addresses the points raised during the review process.

All three Reviewers have requested major changes to the manuscript. Please address all Reviewers’ comments in the revised version of your manuscript.

We look forward to receiving your revised manuscript.

Kind regards,

Benedetta Ruzzenente

Academic Editor

PLOS ONE

Journal Requirements:

3. Please ensure that you refer to Figure 7 in your text as, if accepted, production will need this reference to link the reader to the figure.

Reviewers' comments:

Reviewer's Responses to Questions

**Comments to the Author**

1. Is the manuscript technically sound, and do the data support the conclusions?

Reviewer #1: Partly

Reviewer #2: Partly

Reviewer #3: Partly

2. Has the statistical analysis been performed appropriately and rigorously? 

Reviewer #1: Yes

Reviewer #2: No

Reviewer #3: No

3. Have the authors made all data underlying the findings in their manuscript fully available?

Reviewer #1: Yes

Reviewer #2: Yes

Reviewer #3: No

4. Is the manuscript presented in an intelligible fashion and written in standard English?

Reviewer #1: Yes

Reviewer #2: No

Reviewer #3: No

5. Review Comments to the Author

Reviewer #1: The present study in icefish (the only vertebrate species devoid of haemoglobin) explored the mitochondrial sequence variation for ATP synthase subunit a (ATP6) and b (ATP8). The protein structures were compared with other seven vertebrate species in order to reveal important physiological differences, contributing to understand the unique biology of icefish. Therefore, the study is original and with a high potential interest, but not clear conclusions are stated in the Abstract and/or Discussion sections. This is perhaps the main criticism, and the authors should try to achieve clearer outputs, conducting some physiological validation if it is needed. In the present form, the study is too much descriptive and it is no clear how it contributes to generate new knowledge.

In previous studies in fish model species and farmed fish (including salmonids and non salmonids), the specific regulation of enzyme subunits of the mitochondrial respiratory chain has been addressed in response to changes in food availability and other stressors. These and other physiological studies in other vertebrates should be considered to improve the Discussion section.

The study is focused on ATP6 and ATP8, but why not other enzyme subunits of the respiratory chain encoded by the mitochondrial or nuclear DNA. A more ambitious study, addressing changes in enzyme subunits of the five enzyme complexes of the mitochondrial respiratory chain should be considered.

Reviewer #2: General comment

The article opens with a relatively unclear goal: “shed light on the molecular evolution of these proteins in vertebrate species,” meant ATP synthase F0 subunit components coded by mtATP6 and mtATP8. This goal doesn’t seem to correspond with the manuscript title so much. The Methods don’t lead to this goal so clearly and the goal remains reduced in Discussion to:

1. an unclear statement: “The initial difference we see is that red-blooded species N. coriiceps, E. maclovinus, T. bernacchii, and the two icefish, C. rastrospinosus that are devoid of haemoglobin but have myoglobin and C. aceratus species with a nearly identical gene to that of C. rastrospinosus, have an additional 12 nucleotides at the N-terminal. The only exception to this is the icefish C. gunnari”;

2. relatively nicely written subchapter: “GTG as an alternative start codon”;

3. relatively short, but not so clear subchapter: “Overlap of ATP8 and ATP6 genes”;

4. relatively long, but not so clear subchapter: “Protein Alignment and Structural changes in ATP6”.

Conclusion section is missing.

The authors didn’t generate their own data and they are using published data instead. The authors didn’t develop any new method of data analysis and their results were generated often by easy-to-use web-based software. Majority of the analyses were done on just 13 model vertebrate species. The authors don’t explain so clearly the reason for their model organism selection, which is questionable. The authors picked 8 teleosts, 1 lizard, 4 mammals (but not human); no cartilaginous fish, no lamprey, no hagfish; no tunicate and/or lancelet as an outgroup.

It seems that authors are unsure with their research goal and their results are reflecting that. Many of their results are not even discussed. So, I recommend the authors to articulate clearly their research goal first, then select carefully the best Materials and Methods allowing them to reach their goal, then comment all their results properly in Discussion and provide the reader with a general Conclusion at the end.

Specific comments

1. Abstract, line 27 “The Channichthyidae family “; Introduction, line 56 “Within the Nototheniidae family the subfamily Channichthyinae” and so on. The authors should select the taxonomic system they consider correct and stick to it in the whole article (I suggest following Near et al 2018). The current text is very confusing when authors are using multiple systems in parallel, without any explanation, and they aren’t even testing any hypotheses intended to correct the taxonomic system in use.

2. Introduction, lines 62,63 “sixteen species of the Channichthyinae subfamily” cited sources 1997, 2003. The Channichthyidae family (or Channichthyinae subfamily) covers 16-23 described species in 11 genera. The views of different researchers on the species composition of the taxon differ a lot, especially within the nominotypical genus Channichthys (see https://link.springer.com/article/10.1134%2FS0032945219060079;
https://www.zin.ru/journals/trudyzin/doc/vol_323_4/TZ_323_4_Nikolaeva.pdf). So, I suggest avoiding such explicit controversial statement, because the authors aren’t even testing any hypotheses intended to resolve the situation.

3. In the final paragraph of Introduction, the authors list the species compared and they provide the Latin name twice. I suggest providing the full scientific name just once in this paragraph, but properly instead, e.g., Champsocephalus gunnari Lönnberg, 1905; and short in the rest of their text, e.g., Ch. gunnari. Furthermore, I suggest adding the family/subfamily information there, because the authors describe the differences in notothenioid thermo-adaptation in family/subfamily levels earlier in the Introduction.

4. The information is repeated in the Methods, Results, and Fig 5 caption: “Sequence logos were created using the webserver WebLogo using 4000 vertebrate protein sequences for the protein ATP6 (http://weblogo.threeplusone.com/).” Unfortunately, the authors don’t provide any detail about the 4000 sequences selection criteria. The authors should avoid such redundant repeating of information, but they should provide all relevant information allowing a verification of their result, in the Method section. Anyway, I didn’t notice these results mentioned in the Discussion.

5. I am wondering how the authors selected the model organisms. It is not mentioned in the article and many of these organisms are not individually discussed. Especially in mammals, I would expect the comparison mainly with human, which is avoided. I suggest to clearly explain why the authors selected each of the models in the Method section and explain in Discussion how the results correspond with the mentioned reason for each model selection.

6. Only the protein based dendrogram (Fig. 3) is mentioned in the text (Method and Result sections) but not the DNA based dendrogram (Fig. 4). The protein based dendrogram (Fig. 3) suggests that zebrafish and killifish are phylogenetically related to a lizard, perhaps even more than to notothenioids, which is obviously not true. Fortunately, it’s well established that lizard is not a teleost. Phylogenetic analysis anticipates selection neutrality of the sequence analyzed, which clearly is not the case in the notothenioid ATP synthase F0 subunits. The distance-based NJ dendrograms thus don’t reconstruct the phylogeny of taxa analyzed, but they are reflecting the unique evolutionary history of these sequences under selection in notothenioids instead. It is interesting that the ATP synthase F0 subunits protein and DNA based dendrograms within the Channichthyidae family correspond with each other and the well-established RADseq based phylogenetic analysis of this clade (Near et al 2018), but it doesn’t apply on the whole notothenioid clade anymore. The sentences in Result section describe this insufficiently. Nonetheless, neither these analyses nor phylogenies are mentioned in the Discussion (and Introduction) at all, which leaves me wondering why the authors performed such analyses. The visualization of these dendrograms is also quite confusing and it deserves more polishing in iTOL so the protein based dendrogram (Fig. 3) is visualized equivalently to the DNA based dendrogram (Fig. 4) at least. These dendrograms could be also rooted, because we know the phylogeny in advance.

7. Table 1 and Table 2 are sharing 5 rows, while Table 1 contains 10 rows in total. So, I think that merging these two tables would simplify the reporting and it would be easier to follow.

Reviewer #3: 1- In the introduction it is mentioned that this work is carried out through the use of sequence analysis and secondary structures, however, the analysis carried out was based on tertiary structure, no secondary structure was determined in this work (see: 123-125); 2- The I-TASSER tool was used for the prediction of tertiary structures. However, in this work the quality of the modeled structures is not verified (Ramachandran plot, Z-score, etc.), which raises questions about the results. This is a very critical aspect.;3- Structural comparisons are made at work, however, the way to proceed is not correct. For this analysis, structural protein alignments (superimposed structure) are performed where measurements such as the TM-score and RMSD are determined, which allow establishing functional relationships even when the sequence similarity is low. If two proteins have a low percentage of identity, but have very similar structures, it is likely that they have the same function. For this analysis there are different programs, an excellent one is "RaptorX Structure Alignment Server" to calculate RMSD and TM-score, the software "PyMOL" for RMSD, and the "server TM-score" of the same authors of I-TASSER to calculate the measure with the same name, TM-score.;4- I don't understand what he meant by the fact that the Graphs were produced with MATLAB version R2018b (9.5.0). This analysis is not clear.;5- I think it is not recommended to use multiple programs for alignment because they have different algorithms and usually return discrepant results. In case of using them as was done in this work, a more rigorous discussion of the comparative result would be advisable. There is a lack of uniformity in the format of the writing (two types of letters in many parts of the text), for example, in Figure 3. Please, review all text.

6. PLOS authors have the option to publish the peer review history of their article (what does this mean?). If published, this will include your full peer review and any attached files.

Reviewer #1: No

Reviewer #2: **Yes: **Zdeněk Lajbner

Reviewer #3: **Yes: **Jorge G. Farias, Ph.D.

---

## [Decision Letter · Decision Letter 1]

26 Aug 2021

PONE-D-21-00605R1

Sequence and structure comparison of ATP synthase F0 subunits 6 and 8 in notothenioid fish.

PLOS ONE

Dear Dr. Chakrabarti,

Thank you for submitting your manuscript to PLOS ONE. After careful consideration, we feel that it has merit but does not fully meet PLOS ONE’s publication criteria as it currently stands. Therefore, we invite you to submit a revised version of the manuscript that addresses the points raised during the review process.

I apologize for the delay in sending my decision, but we had to wait a long time for one of the reviewers, who unfortunately was not able to provide us with their comments.

As you will read below the two reviewers have different opinions regarding the acceptance of your article. For this reason, it was important to have also the comments from a third reviewer, but in their absence you should reply to all comments of reviewer 2 who recommended reconsideration of your manuscript following revision.

From my side, I understand your decision to keep the western blot for ATP6 only in the response to the reviewers, because antibodies against the 13 hydrophobic mtDNA encoded proteins are very often not specific. Only western blot with cell extracts from EtBr-treated and untreated cells could verify the specificity of the antibody. Can the authors comment if they plan to measure complex V activity or/and perform a BN-PAGE followed by in gel activity assay to investigate the function of the ATP synthase. Both these approaches could provide more physiological data.

We look forward to receiving your revised manuscript.

Kind regards,

Benedetta Ruzzenente

Academic Editor

PLOS ONE

Reviewers' comments:

Reviewer's Responses to Questions

**Comments to the Author**

1. If the authors have adequately addressed your comments raised in a previous round of review and you feel that this manuscript is now acceptable for publication, you may indicate that here to bypass the “Comments to the Author” section, enter your conflict of interest statement in the “Confidential to Editor” section, and submit your "Accept" recommendation.

Reviewer #1: All comments have been addressed

Reviewer #2: (No Response)

2. Is the manuscript technically sound, and do the data support the conclusions?

Reviewer #1: Partly

Reviewer #2: Partly

3. Has the statistical analysis been performed appropriately and rigorously? 

Reviewer #1: Yes

Reviewer #2: Yes

4. Have the authors made all data underlying the findings in their manuscript fully available?

Reviewer #1: Yes

Reviewer #2: Yes

5. Is the manuscript presented in an intelligible fashion and written in standard English?

Reviewer #1: Yes

Reviewer #2: No

6. Review Comments to the Author

Reviewer #1: This second revised version meets almost all the queries and concerns. Therefore, no addtional comments are needed from the authors.

Reviewer #2: I believe the manuscript reads better now. The authors frequently respond to reviewers’ comments that they addressed the issues within their manuscript. Nonetheless, I think their responses could be more specific and they could address many of the comments within their manuscript even better:

1. I believe that the authors didn’t address my following specific comment (5) in their revision sufficiently: “I am wondering how the authors selected the model organisms. It is not mentioned in the article and many of these organisms are not individually discussed. Especially in mammals, I would expect the comparison mainly with human, which is avoided. I suggest to clearly explain why the authors selected each of the models in the Method section and explain in Discussion how the results correspond with the mentioned reason for model selection on individual level.”

At the end of Introduction (lines 133-136), the authors write that the model organisms were selected “to shed light on the changes of these proteins in the notothenioid species by comparing them to better characterised diverse vertebrate species. These species choices help us decipher amino acid changes specific to notothenioids and those that are potentially species specific (Supplementary Fig 1.).” I believe that this goal could be equally poorly achieved with many of the 5947 ATP6 sequences that the NCBI RefSeq collection offered at the moment of search for WebLogo (I would consider it appropriate to provide also the date of the search). On the other hand, protein sequences from phylogenetically closely related fish groups would allow the authors to address this question appropriately. Hopefully, the authors have another reason for selection of exactly each of these model organisms. They should spell out their reason for each of these model organisms’ selection clearly and discuss their results. What group of animals each of these selected models represent?

I noticed the authors decided to add the human to their selection. It seems meaningful, because they already mentioned human in their manuscript repeatedly anyway. Nonetheless, the addition is not justified in their manuscript either and it is somehow confusing. For example, in Result lines 266-271, it seems the authors are treating human as a species of Antarctic fish.

2. In the specific comment (3) I write: “I suggest providing the full scientific name just once in this paragraph, but properly instead, e.g., Champsocephalus gunnari Lönnberg, 1905; and short in the rest of their text, e.g., Ch. gunnari.”

I am glad the authors don’t repeat the same information twice within the same paragraph anymore. Nonetheless, they are still randomly repeating the genus names, while mentioning the same species in other parts of their manuscript, e.g., Methods 159-163, or Results 291 (Notothenia coriiceps), 292 (Champsocephalus gunnari).

In the same comment, I suggested an addition of a family name to the list to make the species selection easier to follow. Maybe, an addition of a common name and the group of animals or a physiological trait that the model aims to represent would be even better.

3. I am glad the authors add the conclusion section (lines 458-466). Unfortunately, the initial sentence is rather weak, and the authors should consider rephrasing: “In this study we suggest that substitution of hydrophilic amino acid serine (TCT) to hydrophobic amino acid alanine (GCT) in C. gunnari could be a base substitution for thymine to guanine at N1 position of the codon which might have structural impact on the protein.”

4. The authors newly added the sentence to the Abstract: “Though these sequences have been taken from Refseq database validated by different sources it is important to recognise they could still be prone to error.” I think that such information belongs to Methods or maybe Discussion, but not Abstract. None of the reviewers suggested an addition of such information to the Abstract. RefSeq (NCBI) is a commonly used and well established publicly available database. The authors don’t provide any test or a result regarding the database reliability. In case that the authors intend to educate the readers about usefulness of the common resource, they should cite a relevant literature at least, e.g. https://www.ncbi.nlm.nih.gov/pmc/articles/PMC4702849/

7. PLOS authors have the option to publish the peer review history of their article (what does this mean?). If published, this will include your full peer review and any attached files.

Reviewer #1: **Yes: **Jaume Pérez-Sánchez

Reviewer #2: **Yes: **Zdeněk Lajbner

---

## [Author Response · Author response to Decision Letter 1]

1 Sep 2021

Dear Editor, 

Thank you for sending your decision regarding our manuscript, we are submitting a revised version of the manuscript “Sequence and structure comparison of ATP synthase F0 subunits 6 and 8 in notothenioid fish”. We note that reviewer 1 is happy to accept the version already submitted and have addressed the further comments from reviewer 2. 

I am quite astonished by your request for data from newly suggested laboratory experiments testing physiology parameters. I feel that at this stage of a rather delayed review process they are beyond the scope of this particular manuscript, though of course additional data are always interesting. We will duly consider these suggestions in any further work on this subject.

Comments from Reviewer 2 (line numbers are relevant for the ‘clean copy’)

• Comment 1: “I suggest to clearly explain why the authors selected each of the models in the Method section and explain in Discussion how the results correspond with the mentioned reason for model selection on individual level.”

Response: Line numbers-156-160 in methodology addresses the comment. 

• Comment 2: “I noticed the authors decided to add the human to their selection. It seems meaningful because they already mentioned human in their manuscript repeatedly anyway. Nonetheless, the addition is not justified in their manuscript either and it is somehow confusing. For example, in Result lines 266-271, it seems the authors are treating human as a species of Antarctic fish.”

Response: We have now specifically addressed this point, please see Line numbers: 158-160

• Comment 2: “Nonetheless, they are still randomly repeating the genus names, while mentioning the same species in other parts of their manuscript, e.g., Methods 159-163, or Results 291 (Notothenia coriiceps), 292 (Champsocephalus gunnari).

In the same comment, I suggested an addition of a family name to the list to make the species selection easier to follow. Maybe, an addition of a common name and the group of animals or a physiological trait that the model aims to represent would be even better.” 

Response: Suggested changes have been incorporated; Methodology lines- 166-171. Results, lines, 291-292. 

The family name of the species has been added in the Introduction, lines 120-125. The common name of the species is already in Feature Table 1.

• Comment 3: “Unfortunately, the initial sentence is rather weak, and the authors should consider rephrasing: “In this study we suggest that substitution of hydrophilic amino acid serine (TCT) to hydrophobic amino acid alanine (GCT) in C. gunnari could be a base substitution for thymine to guanine at N1 position of the codon which might have structural impact on the protein.”

Response: We have added this suggestion into our concluding paragraph.

• Comment 4: “Though these sequences have been taken from Refseq database validated by different sources it is important to recognise they could still be prone to error.” I think that such information belongs to Methods or maybe Discussion, but not Abstract. None of the reviewers suggested an addition of such information to the Abstract. RefSeq (NCBI) is a commonly used and well established publicly available database. The authors don’t provide any test or a result regarding the database reliability. In case that the authors intend to educate the readers about usefulness of the common resource, they should cite a relevant literature.

Response: We have incorporated reviewer’s suggestion and moved this to the start of the methodology section, line 138 onwards.

We hope these changes now meet all expectations.

Sincerely, 

Lisa Chakrabarti

---

## [Editor Report · Decision Letter 2]

10 Sep 2021

Sequence and structure comparison of ATP synthase F0 subunits 6 and 8 in notothenioid fish.

PONE-D-21-00605R2

Dear Dr. Chakrabarti,

We’re pleased to inform you that your manuscript has been judged scientifically suitable for publication and will be formally accepted for publication once it meets all outstanding technical requirements.

Kind regards,

Benedetta Ruzzenente

Academic Editor

PLOS ONE

Additional Editor Comments (optional):

Thank you for answering to the questions of Reviewer 2.

I just want to point out that the authors misunderstood my comments. I did not request any additional experiments but instead I asked why the author decided to perform a western blot and not more classical physiological experiments in reply to the question of Reviewer 1 concerning the possibility to provide more physiological data. 
---

## [Editor Report · Acceptance letter]

28 Sep 2021

PONE-D-21-00605R2 

*Sequence and structure comparison of ATP synthase F0 subunits 6 and 8 in notothenioid fish*. 

Dear Dr. Chakrabarti:

I'm pleased to inform you that your manuscript has been deemed suitable for publication in PLOS ONE. Congratulations! Your manuscript is now with our production department. 

Kind regards, 

on behalf of

Dr. Benedetta Ruzzenente 

Academic Editor

PLOS ONE